# Speculate Deep and Accurate: Lossless and Training-Free Acceleration for Offloaded LLMs via Substitute Speculative Decoding

Pei-Shuo Wang[1], Jian-Jia Chen*[1], Chun-Che Yang*[1], Chi-Chih Chang[2], Ning-Chi Huang[1], Mohamed S. Abdelfattah[2], and Kai-Chiang Wu[1]

[1]National Yang Ming Chiao Tung University
[2]Cornell University

## Abstract

The immense model sizes of large language models (LLMs) challenge deployment on memory-limited consumer GPUs. Although model compression and parameter offloading are common strategies to address memory limitations, compression can degrade quality, and offloading maintains quality but suffers from slow inference. Speculative decoding presents a promising avenue to accelerate parameter offloading, utilizing a fast draft model to propose multiple draft tokens, which are then verified by the target LLM in parallel with a single forward pass. This method reduces the time-consuming data transfers in forward passes that involve offloaded weight transfers. Existing methods often rely on pretrained weights of the same family, but require additional training to align with custom-trained models. Moreover, approaches that involve draft model training usually yield only modest speedups. This limitation arises from insufficient alignment with the target model, preventing higher token acceptance lengths. To address these challenges and achieve greater speedups, we propose SUBSPEC, a plug-and-play method to accelerate parameter offloading that is lossless and training-free. SubSpec constructs a highly aligned draft model by generating low-bit quantized substitute layers from offloaded target LLM portions. Additionally, our method shares the remaining GPU-resident layers and the KV-Cache, further reducing memory overhead and enhance alignment. SubSpec achieves a high average acceptance length, delivering $9.1\times$ speedup for Qwen2.5 7B on MT-Bench (8GB VRAM limit) and an average of $12.5\times$ speedup for Qwen2.5 32B on popular generation benchmarks (24GB VRAM limit). The code is available at `https://github.com/NYCU-EDgeAi/subspec`.

## 1 Introduction

Large language models (LLMs) [1, 4, 37] have achieved widespread popularity in tasks ranging from chat models to code generation. Local deployment of these models on consumer hardware offers compelling advantages: data privacy, potential cost reductions compared to API access, freedom for model customization, and direct control over the inference process [47].

The substantial memory requirement is the primary barrier to such local deployment. Popular open source model families like Llama [15, 39], Qwen [42, 43], and DeepSeek [26] often exceed the memory constraint in common consumer-level GPUs (typically ranging from 8GB to 24GB). For

---

*These authors contributed equally to this work

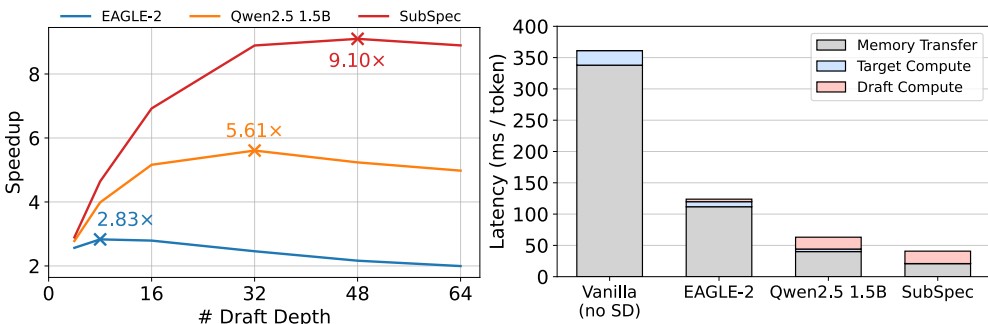

Figure 1: Impact of draft model characteristics on speculative decoding performance, tested under the MT-Bench [46] benchmark. **Left:** Maximal speedup achieved by different draft models varies with draft depth in tree-based speculative decoding. **Right:** Average inference latency per token for Qwen2.5 7B (8GB GPU memory constraint) of different methods with optimal draft depths. SubSpec utilizes a higher draft model computation to minimize costly memory transfers of target model parameters.

instance, Llama3.1 8B requires 16GB VRAM, surpassing the capacity of a consumer card such as an RTX 3060 (12GB).

A common strategy to accommodate LLMs within limited memory is model compression, primarily via techniques like quantization [13, 25]. Quantization reduces model memory demands by encoding weights in low-precision formats (e.g., INT8, INT4); however, this lossy compression inevitably alters model outputs and impairs model quality.

Parameter offloading presents a lossless alternative, storing inactive layers in host memory and streaming them to the GPU when needed [20]. Unfortunately, frequent data transfers over the PCIe bus severely throttle throughput. This results in prohibitively long latencies for token generation, often only one to two tokens per second on consumer cards like the RTX 4090 supporting PCIe 4.0. Such latency undermines practical usability and fails to satisfy demands for real-time interaction.

Speculative decoding (SD) has emerged as a promising acceleration technique to mitigate the excessive latency due to parameter offloading. SD utilizes a smaller, faster draft model to rapidly propose multiple tokens, which are then verified by the larger target model in parallel using a single forward pass. Potentially accepting multiple tokens, SD can substantially reduce the number of expensive forward passes involving target model weight transfers. However, some existing SD approaches [10, 35] achieve notable speedups by relying on smaller pretrained models from the same family that align well with the target LLM. These strategies cannot be directly applied to custom-trained models and require additional training for alignment. Other techniques involving draft model training [24, 45] often yield relatively modest speedups when applied on offloading scenarios. This limitation stems from insufficient alignment, leading to average acceptance lengths generally below seven tokens.

Hence, we introduce *Substitute Speculative Decoding* (SUBSPEC), a plug-and-play and training-free approach designed to maximize inference throughput for offloaded LLMs on consumer-grade devices. SubSpec constructs a highly aligned draft model by sharing GPU-resident weights and KV-Cache with the target model, while maintaining low-bit substitute weights for the offloaded layers. This innovative design ensures exceptionally close alignment between the draft and target models and enables deployment of large models locally with competitive inference speeds. The effectiveness of this approach in reducing data transfer latency is demonstrated in Figure 1.

In this work, our key contributions are as follows:

- Our analysis identifies the predominance of the alignment and draft depth of the draft model under parameter offloading scenarios, compared to its speed.

- We propose a plug-and-play and training-free method that constructs highly aligned draft models using shared components and low-bit substitutes, minimizing VRAM overhead.

- We introduce refinements to draft tree construction, such as probability sharpening, to effectively leverage deeper trees and boost acceptance length.

- We developed an efficient system to accelerate offloading. Using this system, popular models like Qwen2.5 7B achieve 25 tokens per second on a single consumer GPU across diverse benchmarks (within an 8GB VRAM limit), demonstrating over $10\times$ speedups compared to baseline offloading inference.

## 2 Background

### 2.1 Parameter Offloading

Parameter offloading is a technique commonly employed by inference frameworks [14, 18, 21] to manage models whose memory requirements exceed GPU capacity. This strategy primarily stores model parameters in CPU memory, transferring them to the GPU only when required for computation. After the GPU completes computations for a specific layer, its parameters may be either discarded or overwritten by those required for subsequent layers. Typically, frameworks aim to retain as many model layers as possible directly within GPU memory, offloading only the remainder to minimize data transfer overhead.

However, these frequent parameter transfers between CPU and GPU memory introduce considerable latency, as GPU operations often stall while awaiting data. This latency significantly impacts the total inference time, making performance highly sensitive to PCIe bandwidth limitations.

While techniques like Deepspeed-inference [3] and FlexGen [34] improve throughput via large batches, such approaches are not suitable for latency-critical online inference, where small batch processing is standard. Each forward pass remains constrained by the PCIe bus data transfer bottleneck in such scenarios.

### 2.2 Speculative Decoding

Speculative decoding [7, 22] accelerates the target autoregressive LLM by generating multiple tokens per iteration, rather than just one. In each iteration, a smaller, faster *draft model* quickly produces a set of draft tokens. These draft tokens are then evaluated in parallel by the *target model* (the original LLM being accelerated) with a single forward pass. Tokens confirmed by the target model are then accepted, reducing the total forward passes required to generate the full context. The average acceptance length ($\tau$) is the mean number of accepted draft tokens per iteration.

Miao et al. further advanced this approach with tree decoding. This method improves the number of tokens accepted per iteration while conserving computational efficiency. Tree decoding maintains a hierarchical token structure instead of parallel beams. Multiple branching token paths are flattened and evaluated in one forward pass. Positional encodings and attention masks are modified to preserve tree structure dependency. Subsequent works [6, 10, 23, 24] have built on this foundation. These works developed advanced tree-based speculative decoding strategies, reporting $2\times$ to $4\times$ faster than standard autoregressive decoding when no offloading is required.

Recent methods illustrate the benefits of using SD to accelerate offloading scenarios. SD significantly reduces data transfer overhead by reducing the total number of forward passes required by the target model, without losing quality. For example, SpecExec [35] demonstrates that speculating and verifying with a larger budget (from 128 to 2048 draft tokens) per iteration can achieve higher average acceptance length and additional speedup. In contrast, Dovetail [45] focuses on accelerating smaller LLMs on heterogeneous setups, running Llama-2 7B by offloading partial computation to the CPU. The authors of Dovetail also trained a draft model larger than EAGLE [23] to achieve better alignment. These techniques have showcased notable speedups. Such speedups result from choosing mid-sized, accurate draft models and speculating deeper token trees. These approaches either assume access to a compatible smaller model in the same family or require fine-tuning to align draft and target distributions.

# 3 Analysis of Speculative Decoding Speedup in Offloading Scenarios

## 3.1 Theoretical Speedup Analysis

This subsection derives the theoretical speedup of speculative decoding (SD) over autoregressive decoding (AR). The goal is to clarify the factors governing SD performance and highlight the distinct optimization challenges posed by standard (model fully GPU-resident) versus parameter-offloading inference scenarios.

First, we establish the original time required for the token generation of the AR-based target LLM. The total time $T_{AR}^{\mathcal{N}}$ required to generate $\mathcal{N}$ tokens using autoregressive decoding is directly proportional to the number of tokens, as each token necessitates one forward pass of the target model:

$$T_{AR}^{\mathcal{N}} = \mathcal{N} \cdot t_{Target}, \tag{1}$$

where $t_{Target}$ represents the latency of a single forward pass of the target model.

In contrast, the total time $T_{SD}^{\mathcal{N}}$ to generate $\mathcal{N}$ tokens using speculative decoding (SD) is given by:

$$T_{SD}^{\mathcal{N}} = \mathcal{N} \cdot \frac{\mathcal{D} \cdot t_{Draft} + \gamma \cdot t_{Target}}{\tau}, \quad 1 \leq \tau \leq \mathcal{D} + 1, \tag{2}$$

$t_{Draft}$ is the latency of a single draft model forward pass, and $\mathcal{D}$ is the draft depth. For each iteration, the draft model runs $\mathcal{D}$ forward passes to generate a draft token sequence or tree of depth $\mathcal{D}$ (as to "speculate"). The target model then performs a single forward pass over all draft tokens, checking their correctness (as to "verify"). The factor $\gamma$ reflects the relative cost of this parallel verification compared to a normal AR forward pass $t_{Target}$ (typically ranging from $1 \leq \gamma \leq 2$). The term $\tau$, known as the *average acceptance length*, denotes the mean number of tokens accepted per iteration (potentially including a bonus token derived from the final accepted token, thus $1 \leq \tau \leq \mathcal{D} + 1$).

The theoretical speedup is then the ratio $T_{AR}^{\mathcal{N}}/T_{SD}^{\mathcal{N}}$. Combining Equations (1) and (2) yields:

$$\frac{T_{AR}^{\mathcal{N}}}{T_{SD}^{\mathcal{N}}} = \frac{\tau \cdot t_{Target}}{\mathcal{D} \cdot t_{Draft} + \gamma \cdot t_{Target}} = \frac{\tau}{\frac{\mathcal{D} \cdot t_{Draft}}{t_{Target}} + \gamma}, \quad 1 \leq \tau \leq \mathcal{D} + 1. \tag{3}$$

Equation (3) reveals a key trade-off when selecting the draft model and its draft depth ($\mathcal{D}$). In standard settings, both the target latency $t_{Target}$ and the speculation overhead $\mathcal{D} \cdot t_{Draft}$ impact the denominator. Increasing draft depth ($\mathcal{D}$) to improve $\tau$ potentially must therefore be balanced against this linear rise in speculation overhead. This balance typically favors smaller, faster draft models ($t_{Draft} < t_{Target}$) with moderate $\mathcal{D}$, though such models often offer lower alignment with the target, potentially capping the achievable $\tau$.

Conversely, this optimization landscape changes dramatically in parameter offloading scenarios where data transfers significantly increase $t_{Target}$. Here, the relative impact of the speculation latency ($\mathcal{D} \cdot t_{Draft}$) diminishes against the target verification cost ($\gamma \cdot t_{Target}$). Maximizing the average acceptance length ($\tau$) thus becomes crucial to minimize the frequency of expensive target model forward passes. This priority favors draft models with superior target alignment and contextual quality, even if their $t_{Draft}$ is larger, as reducing calls to the costly target model results in greater overall speedup.

## 3.2 Empirical Validation and Motivation for Efficient Draft Model Generation

To empirically demonstrate these contrasting dynamics of speculative decoding (SD) in standard versus parameter-offloading scenarios, we present an illustrative evaluation with Qwen2.5 7B as the target model. Figure 2 showcases the performance of two representative existing draft model types: EAGLE-Qwen2.5* [23], a smaller, generally faster draft model, and Qwen2.5 1.5B [43], a moderately-sized model from the same family that might offer better intrinsic alignment. Figure 2 also provides an early glimpse of SubSpec, our proposed method, which will be detailed in the next section.

---

*The draft model weights for Qwen2.5 were obtained from https://huggingface.co/leptonai/EAGLE-Qwen2.5-7B-Instruct.

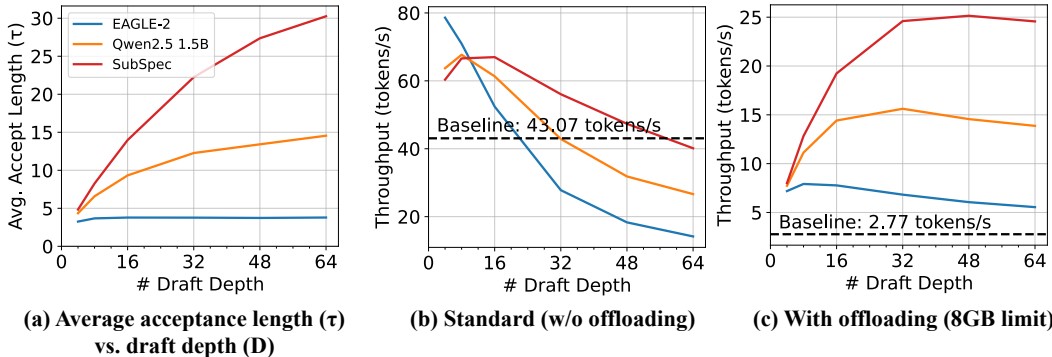

(a) Average acceptance length (τ) vs. draft depth (D)  (b) Standard (w/o offloading)  (c) With offloading (8GB limit)

Figure 2: Comparative performance of Qwen2.5 7B using SD with different draft approaches across varying draft depths, tested under the MT-Bench benchmark.

The results presented in Figure 2 confirm our theoretical speedup analysis (Equation 3): the faster EAGLE-Qwen2.5 draft model performed best in standard settings due to its low overhead, while the better-aligned Qwen2.5 1.5B model achieved superior speedup in the offloading setting, where large $t_{Target}$ dominates due to obtaining a higher average acceptance length ($\tau$). These empirical findings highlight a critical challenge: maximizing SD benefits in offloading scenarios demands highly aligned draft models. This then raises a practical question for us: **How can we efficiently obtain such highly aligned draft models, particularly for custom or fine-tuned LLMs?**

While some open-source model families offer pretrained models of various sizes, allowing smaller versions to serve as potentially efficient and aligned draft models, this option is often unavailable for custom-trained or fine-tuned LLMs. Creating a sufficiently aligned draft model for such custom models typically necessitates additional training or distillation. These processes introduce further costs (computation, memory, data, and time), preventing widespread SD adoption in many real-world deployments.

To bridge this gap, we propose a practical and efficient alternative: *utilizing a low-bit quantized version of the target LLM itself as the draft model, which remains fully GPU-resident*. This approach capitalizes on efficient data-free quantization techniques, eliminating the need for training datasets and resource-intensive training runs. Detailed later in Section 4, our method also incorporates weight and KV-Cache sharing, significantly reducing VRAM overhead while enhancing draft model alignment.

## 4 Substitute Speculative Decoding (SubSpec)

We introduce *Substitute Speculative Decoding (SubSpec)*, a novel method enabling efficient LLM inference on consumer-grade GPUs, particularly when model weights exceed available GPU memory. SubSpec achieves this by constructing highly aligned, fully GPU-resident draft models and performing tree-based speculative decoding, constructing deep draft trees with optimizations.

SubSpec constructs the draft model with quantized 'substitute' layers for the offloaded portions of the target model, while GPU-resident layers and the KV-Cache are shared between the draft and target. This design creates a draft model that is highly aligned with the target model. By constructing deep draft token trees with such a model, it obtains extremely high average acceptance lengths ($\tau$), significantly reducing the expensive data transfers, boosting overall inference throughput. The subsequent sections detail the draft model construction (Section 4.1), adaptations for draft tree construction (Section 4.2), and complementary performance optimizations (Section 4.3).

### 4.1 Draft Model Construction

SubSpec employs three synergistic strategies to ensure that the draft model remains entirely on the GPU. These strategies involve using substitute weights, sharing GPU-resident layers, and employing a shared KV-Cache, as illustrated in Figure 3.

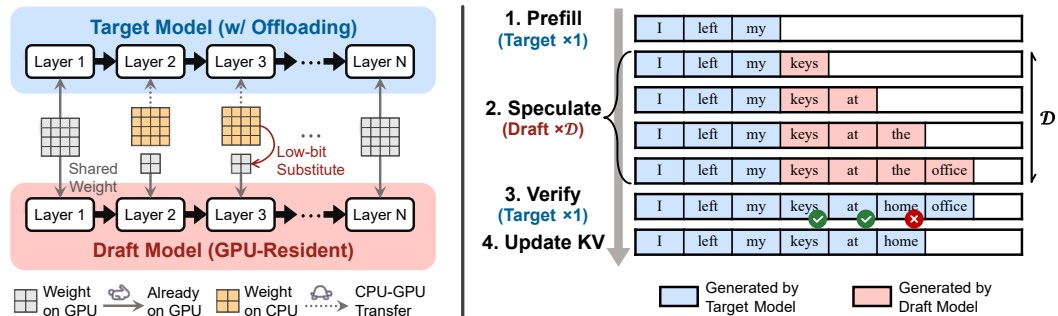

Figure 3: **Left:** Draft model architecture of SubSpec. SubSpec maintains additional low-bit substitute weights to keep the full draft model on the GPU. **Right:** Shared KV-Cache generation pipeline of SubSpec. The draft model reuses the KV-Cache of the target model to achieve better alignment and memory efficiency. This illustration serves as a simple sequential demonstration, while in practice, we maintain a flattened token tree for tree decoding.

**Quantized Substitute Weights for Offloaded Layers.** A core principle of SubSpec involves replacing the offloaded layers of the target model with lightweight, low-bit 'substitute' layers within the draft model. These substitutes reside entirely on the GPU and approximate the functionality of their corresponding target layers. For example, layers 2 and 3 on the left side of Figure 3 of the target model are offloaded and the corresponding low-bit substitutes of the target layers are utilized for the GPU-resident draft model. We generate these substitutes using fast, data-free quantization methods (e.g., HQQ [5], HIGGS [28]), which require minimal processing time (under minutes for 7B to 70B parameter models on a single consumer GPU). During inference, these quantized layers enable rapid execution through highly optimized low-bit GEMM kernels [5, 16].

**GPU-Resident Layer Sharing.** The draft model reuses target model layers that remain in GPU memory. For example, the weight of layer one on the left side of Figure 3 is GPU-resident and shared by both the target model and our draft model. This sharing strategy maximizes GPU resource utilization and inherently improves the alignment between the draft and target models, given that identical weights are employed for these shared layers.

**Shared KV-Cache.** The structural similarity between the draft model of SubSpec and the target model allows for a unified KV-Cache. This sharing approach yields significant advantages: it halves the KV-Cache memory footprint compared to using separate caches and enhances alignment by ensuring that both models operate on an identical contextual history. Furthermore, sharing the KV-Cache eliminates the need for a distinct prefilling phase for the draft model, directly contributing to faster overall inference. The demonstration of this pipeline is illustrated on the right side of Figure 3. The draft model extends new KV-Cache values on speculation, which the target model then overwrites during verification to ensure identical results.

### 4.2 Optimized Draft Tree Construction

**Constructing Deep Context-Aware Dynamic Draft Tree.** The high alignment of the SubSpec draft model (detailed in Section 4.1) enables the exploration of deeper draft trees to achieve higher average acceptance lengths ($\tau$). Our sampling approach thus extends established context-aware dynamic draft tree methods like EAGLE-2 [24] and SpecExec [35] to support these greater depths.

A context-aware dynamic draft tree is built by iteratively generating draft tokens with the draft model over $\mathcal{D}$ future time steps. In each of these $\mathcal{D}$ forward passes, all leaf nodes are input to the draft model, each yielding probability distributions for the potential next tokens. The score for each potential next token is the cumulative product of its conditional generation probability (from the draft model) and its parent path score. The top-k tokens with the highest scores are selected to form the new $k$ leaf nodes for the subsequent time step. This iterative procedure produces a draft tree of depth $\mathcal{D}$, presenting $k \times \mathcal{D}$ draft tokens for target model validation (not including the root token).

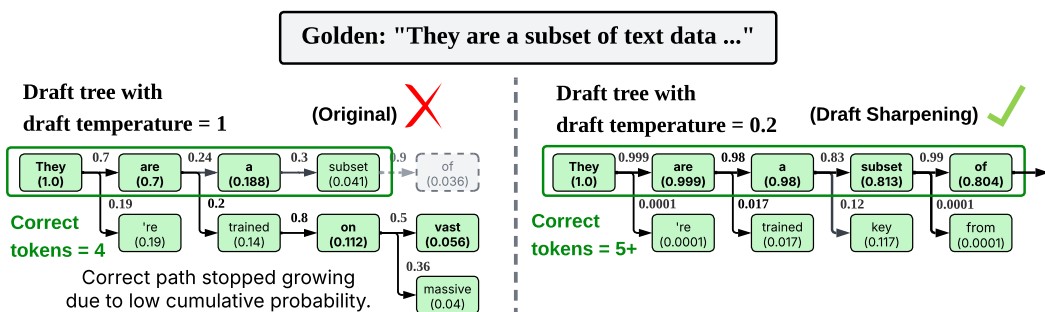

Figure 4: Demonstration of the false positive path issue (with k = 2 for simplification). Correct draft tokens may be dropped due to a lower cumulative probability score. The number in the parentheses in each token denotes the cumulative probability score.

**Addressing Cumulative Probability Divergence.** While exploring substantially deeper draft trees than typically used in prior work (often $\mathcal{D} \leq 7$) showed promise for increasing $\tau$, construction for greedy decoding (target temperature $= 0$) revealed a subtle issue. We observed that paths initiated by less probable tokens could accumulate an erroneously high overall likelihood through high-probability subsequent selections. This phenomenon can lead to 'false-positive' paths that achieve a higher cumulative probability than the genuinely optimal path (illustrated in Figure 4). Such divergence is problematic in greedy decoding, as it can misguide path selection, potentially causing early termination of the correct sequence and thereby limiting the achievable $\tau$.

**Draft Probability Sharpening.** To counteract this divergence under greedy decoding, we employ draft probability sharpening. This technique involves applying a low temperature (= 0.2) to the output distribution of the draft model before calculating cumulative probabilities for tree sampling. Such sharpening makes the probability distribution more peaked, reducing the probability mass allocated to tokens with lower initial probabilities. Further analysis is shown in Appendix D.

## 4.3 Complementary Performance Optimizations

Beyond the core SubSpec framework for draft construction and sampling, two complementary techniques are integrated to further enhance performance and efficiency:

**Asynchronous Data Transfer.** We mitigate the computation cost of the target layers by overlapping computation with data movement. While the GPU processes the current layer, the parameters for the next required offloaded layer are concurrently prefetched from CPU memory. Unlike some prefetching strategies limited to adjacent layers within the same decoder block, our implementation operates across decoder layers, maximizing potential computation hiding. Furthermore, these prefetched layers are loaded into the same reused memory regions to avoid increasing peak memory usage. This asynchronous data transfer technique effectively conceals the computation time of the SD verification step during forward passes of large draft trees.

**Chunked Prefill for Long Contexts.** Prefilling long input prompts can lead to substantial peak GPU memory usage because intermediate activation sizes scale with context length, restricting the number of target model layers that can remain GPU-resident. To address this, we employ chunked prefill, where the input is segmented into fixed-length chunks (e.g., 256 tokens), and the KV-Cache is built incrementally. This method significantly reduces peak memory during the prefilling phase. While Sarathi-Serve [2] introduced this approach primarily to minimize pipeline bubbles in token serving, our adaptation specifically focuses on curtailing peak memory of the prefilling phase for long contexts to maximize the GPU residency of target model layers, further enhancing overall inference efficiency.

Table 1: Throughputs and average acceptance lengths $\tau$ by using different draft models. L31 represents Llama-3.1-Instruct, L32 represents Llama-3.2-Instruct, Q represents Qwen-2.5-Instruct. None represents vanilla autoregressive decoding is used. The number in each parentheses below the target model name denotes the restricted VRAM limit.

| Target | Draft | MT-Bench | | HumanEval | | GSM8K | | Alpaca | | CNN/DM | | Mean | |
|---|---|---|---|---|---|---|---|---|---|---|---|---|---|
| | | tokens/s | $\tau$ | tokens/s | $\tau$ | tokens/s | $\tau$ | tokens/s | $\tau$ | tokens/s | $\tau$ | tokens/s | $\tau$ |
| | | | | | | | Temperature = 0 | | | | | | |
| L31 8B (8GB) | None | 2.40 | 1 | 2.39 | 1 | 2.40 | 1 | 2.40 | 1 | 2.34 | 1 | 2.39 (1.00×) | 1 |
| | EAGLE-2 | 7.56 | 3.90 | 8.58 | 4.43 | 8.05 | 4.15 | 7.59 | 3.90 | 6.45 | 3.34 | 7.65 (3.20×) | 3.95 |
| | L32 1B | 15.14 | 11.91 | 25.79 | 20.23 | 20.17 | 15.71 | 14.98 | 11.54 | 10.05 | 8.66 | 17.23 (7.22×) | 13.61 |
| | SubSpec | **24.29** | **28.35** | **28.09** | **31.63** | **27.78** | **31.55** | **22.13** | **25.38** | **22.60** | **31.39** | **24.98 (10.47×)** | **29.66** |
| Q 7B (8GB) | None | 2.77 | 1 | 2.77 | 1 | 2.77 | 1 | 2.77 | 1 | 2.34 | 1 | 2.68 (1.00×) | 1 |
| | EAGLE-2 | 8.00 | 3.73 | 8.76 | 4.08 | 8.28 | 3.85 | 7.47 | 3.47 | 6.47 | 3.02 | 7.80 (2.91×) | 3.63 |
| | Q 1.5B | 15.78 | 12.27 | 28.29 | 21.88 | 22.86 | 17.59 | 12.37 | 9.47 | 9.34 | 7.88 | 17.73 (6.61×) | 13.82 |
| | SubSpec | **25.35** | **27.08** | **33.48** | **34.77** | **33.04** | **34.18** | **22.30** | **23.19** | **21.28** | **26.33** | **27.09 (10.10×)** | **29.11** |
| Q 14B (12GB) | None | 1.22 | 1 | 1.22 | 1 | 1.22 | 1 | 1.22 | 1 | 1.17 | 1 | 1.20 (1.00×) | 1 |
| | Q 1.5B | 8.17 | 10.81 | **16.20** | 21.45 | 12.58 | 16.54 | 6.46 | 8.43 | 4.40 | 6.40 | 9.56 (7.92×) | 12.73 |
| | SubSpec | **12.05** | **26.36** | 15.70 | **33.76** | **15.44** | **32.94** | **10.34** | **21.79** | **9.27** | **24.08** | **12.56 (10.4×)** | **27.79** |
| Q 32B (24GB) | None | 0.52 | 1 | 0.52 | 1 | 0.52 | 1 | 0.52 | 1 | 0.50 | 1 | 0.52 (1.00×) | 1 |
| | Q 7B | 3.68 | 12.55 | 6.09 | 20.73 | 5.30 | 17.91 | 3.20 | 10.74 | 2.15 | 8.32 | 4.08 (7.86×) | 14.05 |
| | Q 1.5B | 4.49 | 10.87 | **8.22** | 19.90 | 6.74 | 16.17 | 3.61 | 8.62 | 2.39 | 6.46 | 5.09 (9.80×) | 12.40 |
| | SubSpec | **6.33** | **27.53** | 7.58 | **32.70** | **7.96** | **33.66** | **5.80** | **24.50** | **4.80** | **26.48** | **6.50 (12.50×)** | **28.97** |
| | | | | | | | Temperature = 0.6 | | | | | | |
| L31 8B (8GB) | None | 2.40 | 1 | 2.39 | 1 | 2.40 | 1 | 2.40 | 1 | 2.34 | 1 | 2.39 (1.00×) | 1 |
| | EAGLE-2 | 7.38 | 3.81 | 8.30 | 4.29 | 7.60 | 3.92 | 7.49 | 3.86 | 6.19 | 3.21 | 7.39 (3.10×) | 9.96 |
| | L32 1B | 12.30 | 9.69 | 21.37 | 16.82 | 15.88 | 12.42 | 12.80 | 9.83 | 8.16 | 6.93 | 14.10 (5.91×) | 11.14 |
| | SubSpec | **14.62** | **17.58** | **22.59** | **26.04** | **16.51** | **19.54** | **13.17** | **15.46** | **11.61** | **15.37** | **15.70 (6.58×)** | **18.80** |
| Q 7B (8GB) | None | 2.77 | 1 | 2.77 | 1 | 2.77 | 1 | 2.77 | 1 | 2.34 | 1 | 2.68 (1.00×) | 1 |
| | EAGLE-2 | 7.42 | 3.45 | 8.49 | 3.96 | 8.07 | 3.76 | 6.47 | 3.01 | 5.69 | 2.66 | 7.23 (2.69×) | 3.37 |
| | Q 1.5B | 13.19 | 10.43 | 22.96 | 18.06 | 19.19 | 15.02 | 10.59 | 8.21 | 7.63 | 6.44 | 14.71 (5.48×) | 11.63 |
| | SubSpec | **15.92** | **17.09** | **29.13** | **30.57** | **23.89** | **25.32** | **14.43** | **14.98** | **10.15** | **12.07** | **18.70 (6.97×)** | **20.00** |
| Q 14B (12GB) | None | 1.22 | 1 | 1.22 | 1 | 1.22 | 1 | 1.22 | 1 | 1.17 | 1 | 1.20 (1.00×) | 1 |
| | Q 1.5B | 6.53 | 8.62 | 11.90 | 15.73 | **9.87** | 12.99 | 5.43 | 7.09 | 3.77 | 5.42 | 7.50 (6.21×) | 9.97 |
| | SubSpec | **6.90** | **15.22** | **12.17** | **26.74** | 9.56 | **21.02** | **6.14** | **13.05** | **4.40** | **10.90** | **7.83 (6.49×)** | **17.39** |
| Q 32B (24GB) | None | 0.52 | 1 | 0.52 | 1 | 0.52 | 1 | 0.52 | 1 | 0.50 | 1 | 0.52 (1.00×) | 1 |
| | Q 7B | 2.64 | 9.08 | 4.88 | 16.67 | 3.90 | 13.24 | 2.49 | 8.37 | 1.76 | 6.61 | 3.13 (6.03×) | 10.79 |
| | Q 1.5B | 3.71 | 9.03 | 5.72 | 13.93 | **4.80** | 11.61 | 2.94 | 7.06 | 1.94 | 5.26 | 3.82 (7.35×) | 9.38 |
| | SubSpec | **3.74** | **16.40** | **6.15** | **26.54** | 4.41 | **19.32** | **3.37** | **14.16** | **2.65** | **13.33** | **4.06 (7.82×)** | **17.95** |

# 5 Performance Evaluation

## 5.1 Evaluation Setup

**Evaluation Benchmarks.** We evaluated performance across five diverse generative tasks, consistent with the benchmarks from EAGLE [23] and Spec-Bench [41]. These tasks included multi-turn conversation (MT-Bench [46]), code generation (HumanEval [9]), mathematical reasoning (GSM8K [11]), instruction following (Alpaca [36]), and summarization (CNN/Daily Mail [31]).

**Hardware and Simulated Environments.** All experiments were run on a system with an RTX 4090 GPU, an Intel i7-13700K CPU, a PCIe-4.0 x16 bus, and 128GB of DDR5 RAM. GPU memory utilization during evaluations was programmatically restricted to 8GB, 12GB, and 24GB VRAM capacities to simulate diverse consumer device environments.

**Comparative Methodology and Parameters.** We compared the end-to-end speedup of SubSpec against EAGLE-2 [†] and chat models from the Qwen2.5 (7B, 14B, 32B) and Llama3.1 (8B) families. Evaluations used a batch size of 1 under both greedy (target temperature = 0) and stochastic (target temperature = 0.6) generation. For fair comparison, all methods used an identical context-aware dynamic draft tree algorithm without additional tree pruning techniques. A portion of the target model decoder layers was kept resident on the GPU within the VRAM limits. All SD methods were evaluated on 20 identical samples, randomly selected from each dataset. The baseline (offloading with no SD) used the initial five samples due to its significantly longer runtime.

The key parameters were configured as follows: The low-bit substitute layers in SubSpec were quantized to 4 bits with a group size 64 using HQQ. EAGLE-2 used its default published parameters

---

[†]The draft model weights for Llama3.1 were obtained from their official repository.

($k = 10$, $\mathcal{D} = 6$). For SubSpec and the smaller pretrained draft models, the top-k value of tree construction was set to $k = 6$. Their optimal draft depths ($\mathcal{D}$), identified through the grid search reported in Section 3.2 (results shown in Figure 2), were $\mathcal{D} = 48$ for SubSpec and $\mathcal{D} = 32$ for the pretrained models. While further parameter tuning might yield additional improvements, such exhaustive optimization was beyond the scope of this research. Chunked prefill was also applied to prevent out-of-memory (OOM) errors and maximize the number of decoder layers on the GPU.

To better reflect typical real-world usage, all draft models were executed using `torch.compile` with the `max-autotune` configuration. A static KV-Cache with a context length of 2048 tokens was applied consistently across all methods. The comparative results are summarized in Table 1.

## 5.2 End-to-end Performance

The results in Table 1 demonstrate the effectiveness and robustness of SubSpec. SubSpec consistently achieved average acceptance lengths ($\tau$) near 30 across tasks, delivering the highest average throughput. This performance translates to a speedup of $10\times$ to $12.5\times$ between different model sizes, underscoring the broad applicability and significant performance gains offered by SubSpec. We also evaluated the performance of the reasoning models on additional reasoning benchmarks listed in the Appendix G.

Further highlighting its efficiency, SubSpec achieves an additional 30% to 50% speedup compared to smaller draft models from the same family as the target model, without any additional training. This advantage underscores the critical role of the enhanced draft alignment of SubSpec in accelerating offload scenarios. All SD methods showed reduced performance in stochastic settings (target temperature = 0.6). For SubSpec, this meant a decrease in speedup of approximately 60%. Despite this, SubSpec maintained a considerable speedup of $5.8\times$ to $7.8\times$, showcasing its resilience and sustained effectiveness even under less favorable generation conditions. We achieved a low standard deviation of 0.101 tokens/sec on five independent runs of SubSpec on MT-Bench benchmark.

## 5.3 Ablation Study

Finally, we performed an ablation study on MT-Bench under an 8GB VRAM constraint to assess the impact of individual SubSpec components. The results are detailed in Table 2. A baseline configuration of only implementing the core concept of SubSpec (a quantized, GPU-resident draft model using only 'substitute' layers for offloaded portions and shared GPU-resident target model layers), achieved a $7.05\times$ speedup (19.54 tokens/s).

Table 2: Ablation study of SubSpec component contributions for accelerating Qwen2.5 7B target model on MT-Bench under greedy decoding (8GB VRAM limit). Performance is shown as the components were added cumulatively. The final row represents the complete SubSpec system.

| Method | tokens/s | $\tau$ |
|---|---|---|
| Substitute and layer sharing | 19.54 (7.05$\times$) | 23.07 |
| + Shared KV-Cache | 21.99 (7.94$\times$) | 25.14 |
| + Draft prob. sharpening | 23.66 (8.54$\times$) | 27.08 |
| + Async data transfer | 25.35 (9.15$\times$) | 27.08 |

Sequentially integrating additional enhancements of shared KV-Cache, draft probability sharpening, and asynchronous data transfer yielded further performance gains. Each of these components contributed an approximate 7% to 13% increase in throughput. The complete SubSpec system, incorporating all optimizations, ultimately delivered a $9.15\times$ speedup and a throughput of 25.35 tokens/s. These results affirm the individual and collective efficacy of the components of SubSpec.

## 6 Conclusion

This paper addressed the challenge of efficiently performing the inference of large language models on memory-constrained consumer GPUs using parameter offloading. Our analysis confirmed that a highly aligned draft model is crucial for speculative decoding to accelerate parameter offloading effectively. We introduced SubSpec, a novel lossless and training-free technique based on this insight. SubSpec constructs an aligned draft model by utilizing low-bit substitute layers for offloaded portions of the target LLM while sharing GPU-resident components. Evaluations demonstrate that SubSpec is robust across various model sizes and benchmarks under realistic memory limits, achieving substantial average speedups of $10\times$ to $12.5\times$ compared to baseline offloading inference. These results significantly advance the feasibility of deploying large, high-quality LLMs locally on widely available consumer hardware.

## Acknowledgments

The authors would like to thank ASUS for their generous support in providing the computing resources necessary for this work.

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

## A  Limitations

**Minimum GPU Memory Requirement.**  SubSpec has a higher minimum GPU memory requirement. A general prerequisite for speculative decoding to work effectively is that the entire draft model must be GPU-resident. SubSpec architecture necessitates maintaining low-bit substitutes for offloaded target model layers. An approximate 7.1GB minimum GPU memory is required for Qwen2.5 7B, thus fewer layers of the target model can remain GPU-resident compared to some alternative speculative decoding methods. Despite this, SubSpec demonstrates superior throughput in all benchmarks.

**Quantization Granularity.**  Our research only experimented with 4-bit quantization for substitute layers. Although this might affect draft model alignment, more aggressive methods (e.g., 2-bit or 3-bit quantization) could further reduce VRAM demands. Such VRAM savings could permit strategic memory reallocation, for instance, retaining critical draft or target model layers at full GPU precision. A thorough exploration of these trade-offs between draft quality, VRAM usage, and performance represents an important direction for future research.

**Applicability to Model Architectures.**  SubSpec mainly suits dense LLM architectures. Applying SubSpec to alternative architectures, such as Mixture-of-Experts (MoE) models, requires further adaptation and research.

## B  Extended Discussion

### B.1  Comparison with Standalone Quantized Models

While a standalone 4-bit quantized model may fully fit on a GPU and eliminate the offloading bottleneck, this approach inevitably alters outputs and degrades model quality. In contrast, SubSpec avoids this accuracy trade-off entirely.

SubSpec makes parameter offloading practical. As shown in Table 3, this method accelerates naïve offloading from an unusable 2.4 tokens/sec to an acceptable 24 tokens/sec for interactive use on consumer-grade hardware. We therefore position Sub-Spec as a simple, training-free solution for users who refuse to compromise on the model quality.

Table 3: Throughput comparison of various quantization methods for the Llama3.1 8B model on the MT-Bench (8GB VRAM). [§]

| Model Configuration | tokens/s |
|---|---|
| Original (fp16) | 2.40 |
| GPTQ [13] (int4) | 58.87 |
| AWQ [25] (int4) | 52.32 |
| HQQ [5] (int4) | 135.84 |
| SubSpec | 24.29 |

### B.2  Why Tree-Based Speculative Decoding

This research focuses on tree-based speculative decoding (SD) because tree-based SD generally achieves higher average acceptance lengths ($\tau$) than sequential or self-speculation methods [23, 30, 41]. Besides, high $\tau$ values are crucial to minimize expensive forward passes of the target model, especially in offloading scenarios that involve significant data transfer overhead.

### B.3  Future Outlook and Technological Advancements.

Forthcoming interconnect advancements (e.g., PCIe 5.0 and 6.0), along with the continuous progress on model compression methods and kernel optimizations, are anticipated to further enhance inference performance. Each PCIe generation roughly doubles raw bandwidth, halving expensive target model data transfers. While faster PCIe reduces target model latency, potentially increasing the relative cost of speculation iterations, concurrent GPU computation and kernel efficiency improvements are expected to accelerate speculation proportionally. Consequently, SubSpec is projected to retain its relative speedup advantage over standard autoregressive decoding. These combined technological trends promise a progressive reduction in the performance gap between offloaded and non-offloaded LLM inference.

---

[§]For the GPTQ and AWQ methods, we loaded their corresponding pre-quantized weights directly from the Hugging Face Hub.

## C  Related Works

**SpecExec [35].**  SpecExec is a speculative decoding method that introduces a pruning strategy during tree construction and an early-exit mechanism to reduce speculation time.  However, these features increase computational complexity and hinder the application of `torch.compile` due to dynamic shapes from pruning.

We performed a parameter sweep for SpecExec on the MT-Bench benchmark, with results in Table 4. While SpecExec exceeds the 8GB VRAM constraint, SubSpec achieves 76% speedup without requiring an additional draft model.

Table 4: Performance of SpecExec using Llama3.1 8B as target model and Llama3.2 1B as draft model on MT-Bench (greedy decoding). "Budget" denotes the number of tokens concurrently verified, "VRAM" denotes the peak GPU memory.

| Budget | VRAM (GB) | tokens/s | $\tau$ |
|--------|-----------|----------|--------|
| 64 | 8.38 | 11.75 | 8.64 |
| 128 | 8.43 | 12.21 | 9.42 |
| 256 | 8.55 | 13.02 | 10.79 |
| 512 | 8.79 | **13.77** | 12.40 |
| 1024 | 9.27 | 12.95 | 12.93 |

**Dovetail [45].**  Dovetail is a speculative decoding method to accelerate LLM inference on consumer-grade devices by offloading portions of the target computation to the CPU, with the draft model on the GPU. Unlike GPU processing, CPU computation for verification scales linearly with the number of draft tokens. This trait typically restricts the amount of verifiable draft tokens per iteration to maintain optimal speedup. For instance, Dovetail used only 16 draft tokens for verification, whereas our proposed SubSpec verified 288 draft tokens in parallel.

**Self-Speculative Decoding.**  Self-speculative decoding builds a draft model by reusing layers from a target model to reduce parameter overhead and improve alignment. While many approaches add new parameters and require extra training [6, 24, 27, 44], recent works propose training-free layer skipping methods [8, 29, 40]. This distinction highlights an opportunity for hybrid solutions. Future work could explore the combination of SubSpec with layer skipping to produce a draft model with a lower VRAM requirement and faster inference.

## D  Analysis of Draft Probability Sharpening

Table 7 discusses the results of varying draft temperatures among draft model types. The first result (denoted Self in the second row) performs speculative decoding using the target model itself, i.e., Qwen2.5 7B, as the draft model. Although this setup does not make sense from both memory and speedup perspectives, the experiment clearly demonstrates the cumulative probability divergence problem mentioned in Section 4.2, which

Table 5: Average acceptance length ($\tau$) for various draft models and temperatures with Qwen2.5 7B target model on MT-Bench (greedy decoding), where a draft temperature of 1.0 represents the baseline without draft probability sharpening.

| Draft Temp. | 0.2 | 0.4 | 0.6 | 0.8 | 1.0 | 1.2 |
|-------------|-----|-----|-----|-----|-----|-----|
| Qwen2.5 7B (Self) | **34.50** | 33.03 | 31.95 | 29.64 | 27.79 | 26.57 |
| EAGLE-2 | 3.42 | 3.62 | 3.70 | 3.71 | 3.76 | **3.77** |
| Qwen2.5 1.5B | 11.86 | 12.20 | 12.48 | 12.36 | 12.42 | **12.51** |
| SubSpec | 27.08 | **27.19** | 27.16 | 26.09 | 25.29 | 23.92 |

shows that even the same model cannot correctly predict all the tokens when using the default draft tree method.

By lowering the draft temperature, the average acceptance length ($\tau$) increases from 27.79 to 34.50, showing that this aggressive sharpening effectively counters the "false positive" path issue. For SubSpec, $\tau$ also increases from 23.92 to 27.08. This method does not yield a better $\tau$ due to the lower alignment of the pretrained models and EAGLE-2.

# E Verification Pipeline with Asynchronous Data Transfer

Figure 5 illustrates the execution timelines of the verification step in vanilla autoregressive decoding, speculative decoding, and speculative decoding with asynchronous data transfer. The bottom diagram (c) illustrates how asynchronous data transfer works. This method overlaps GPU computation (Q, K) with the data transfer of the subsequent layer (Loading K, V weights to the GPU) to hide the higher verification computation time for speculative decoding.

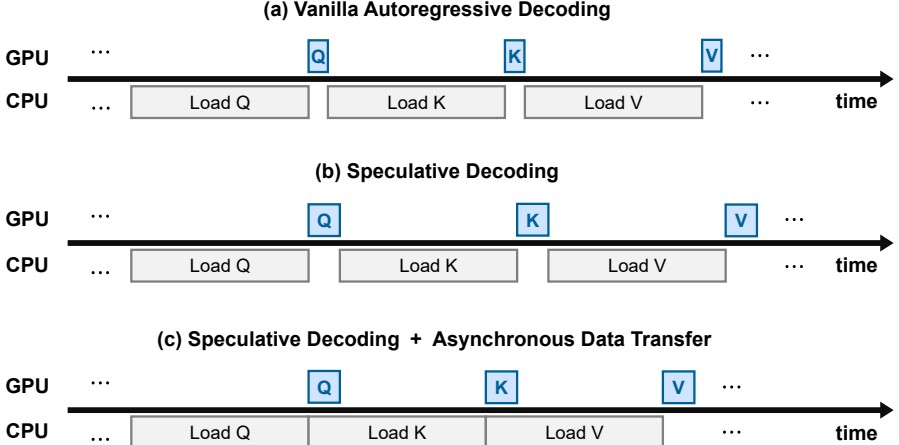

Figure 5: Execution timelines of the verification step for different decoding strategies

# F Temporal Analysis: Speculation vs. Verification

For each iteration in speculative decoding, the draft model runs $\mathcal{D}$ forward passes to generate a draft tree of depth $\mathcal{D}$ (speculate). The target model then performs a single forward pass over all draft tokens, checking their correctness (verify). We compare the average execution time of these two phases, along with the actual obtained throughput, as shown in Figure 6. The optimal throughput for SubSpec occurs at $\mathcal{D} = 48$, where the speculation and verification times are nearly equal.

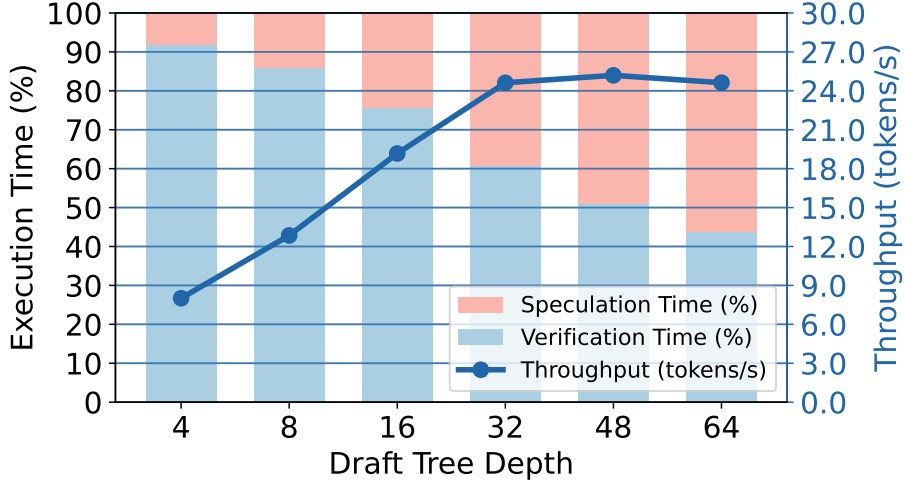

Figure 6: Execution time breakdown (speculation vs. verification) and average acceptance lengths ($\tau$) of SubSpec accelerating Qwen2.5 7B under an 8GB VRAM limit.

## G Performance Evaluation on Reasoning Models

To further demonstrate the robustness of the lossless and training-free method, SubSpec, we also evaluated its performance on reasoning models (DeepSeek-R1 distilled variants [12], QWQ [38]). This evaluation utilized benchmarks widely used in the field: AIME 2024 [32], MATH 500 [17], GPQA Diamond [33], and LiveCodeBench [19]. As shown in Table 6, SubSpec demonstrated promising results of over $10\times$ speedup on all benchmarks.

Table 6: Speedup ratios and average acceptance lengths ($\tau$) for SubSpec compared to baseline (None) on reasoning models under greedy decoding. DSL represents DeepSeek-R1-Distill-Llama, and DSQ represents DeepSeek-R1-Distill-Qwen. The number in each parentheses below the target model name denotes the restricted VRAM limit.

| | | AIME 2024 | | Math 500 | | GPQA Diamond | | LiveCodeBench | | Mean | |
|---|---|---|---|---|---|---|---|---|---|---|---|
| Target | Draft | tokens/s | $\tau$ | tokens/s | $\tau$ | tokens/s | $\tau$ | tokens/s | $\tau$ | tokens/s | $\tau$ |
| DSL 8B | None | 2.40 | 1 | 2.40 | 1 | 2.40 | 1 | 2.39 | 1 | 2.40 (1.00×) | 1 |
| (8GB) | SubSpec | **31.30** | **38.64** | **32.39** | **39.89** | **27.54** | **34.18** | **28.44** | **35.69** | **29.92 (12.49×)** | **37.10** |
| DSQ 7B | None | 2.77 | 1 | 2.77 | 1 | 2.77 | 1 | 2.76 | 1 | 2.77 (1.00×) | 1 |
| (8GB) | SubSpec | **33.45** | **37.76** | **35.25** | **39.55** | **27.74** | **31.41** | **29.94** | **34.45** | **31.60 (11.42×)** | **34.72** |
| DSQ 14B | None | 1.22 | 1 | 1.22 | 1 | 1.21 | 1 | 1.21 | 1 | 1.21 (1.00×) | 1 |
| (12GB) | SubSpec | **17.38** | **40.57** | **17.60** | **40.96** | **14.93** | **35.07** | **15.73** | **37.42** | **16.41 (13.51×)** | **38.51** |
| DSQ 32B | None | 0.52 | 1 | 0.52 | 1 | 0.52 | 1 | 0.52 | 1 | 0.52 (1.00×) | 1 |
| (24GB) | SubSpec | **9.56** | **43.59** | **9.70** | **44.00** | **8.57** | **39.45** | **8.47** | **39.70** | **9.07 (13.76×)** | **41.68** |
| QWQ 32B | None | 0.52 | 1 | 0.52 | 1 | 0.52 | 1 | 0.52 | 1 | 0.52 (1.00×) | 1 |
| (24GB) | SubSpec | **7.95** | **36.29** | **8.11** | **36.93** | **6.86** | **31.54** | **5.89** | **27.47** | **7.20 (13.76×)** | **33.06** |

## H Quantized Target Model Scenarios

We analyzed the practical trade-off between model precision and performance for an 8-bit quantized target model. As detailed in Table 7, this quantization nearly halves the memory footprint and offloading time relative to the 16-bit version. While the process introduces an acceptable loss in accuracy, this configuration increases throughput by 36% with all other configurations held constant. This outcome demonstrates a compelling trade-off, where a minor reduction in precision yields a substantial performance gain.

Table 7: Performance of Llama3.1 8B across different SubSpec settings on MT-Bench (greedy decoding, 8GB VRAM).

| Config (Target, Draft). | tokens/s | $\tau$ |
|---|---|---|
| SubSpec (fp16, int4) | 24.29 | 29.66 |
| SubSpec (int8, int4) | **33.03** | 29.76 |

## I Supplementary Evaluation Configurations

We retain as many target model layers as possible directly within GPU memory to reduce the expensive data transfer overhead. The draft models and KV-Cache are default GPU-resident.

The following details the number of GPU-resident layers of the target model for various methods within VRAM limits. Embedding and head layers are default to be GPU-resident:

- **Vanilla (Baseline Offloading)**: 11 layers for 7B/8B target models, 15 layers for 14B target models, and 20 layers for 32B target models.
- **EAGLE-2**: 7 layers for 7B/8B target models.
- **SD with Mid-Size Pretrained Draft Model (1B/1.5B)**: 3 layers for 7B/8B target models, 7 layers for 14B target models, and 16 layers for 32B target models.
- **SD with Large Pretrained Draft Model (7B)**: 3 layers for the 32B target model.
- **SubSpec**: All decoder layers of the target model were offloaded, with their 4-bit quantized substitutes retained on the GPU.

