# OpenReview forum: "Speculate Deep and Accurate: Lossless and Training-Free Acceleration for Offloaded LLMs via Substitute Speculative Decoding"
_NeurIPS.cc/2025/Conference — NeurIPS 2025 poster_

### Official Review · Reviewer_DA4c · 2025-07-02

**Clarity:** 3
**Significance:** 2
**Originality:** 2
**Rating:** 5
**Confidence:** 3

**Summary:**

This paper presents SubSpec, a speculative decoding method to accelerate inference for LLMs using parameter offloading. Specifically, SubSpec selects a number of layers to be quantized and the rest of layers to be GPU-resident, while sharing the KV cache. This leads to a training-free draft model with strong alignment with the target. The evaluation results show strong acceleration (up to 12.5$\times$) on Qwen2.5 32B, with 24GB VRAM limit.

**Questions:**

- Could you please elaborate on how throughput translates to speedups? especially Tab 1 is very similar to that of Eagle-2, but Eagle-2 reported speedups. It would be helpful to include speedups in the results
- Could you discuss the implementation complexity and system requirements? e.g. incorporating HQQ for quantization, and does the implementation of SubSpec pose any constraints on systems?
- Thank you in advance for any response to the above.

**Ethical Concerns:**

["NO or VERY MINOR ethics concerns only"]

**Final Justification:**

My concerns have been sufficiently addressed, and I am raising my score to Accept.

**Limitations:**

Yes

**Quality:**

2

**Strengths And Weaknesses:**

Strengths

- The objective of improving SD acceleration on resource constrained hardware is well motivated. The paper is well-structured and clearly presented.
- The proposed method of low-bit quantization of selected layers while keeping GPU resident layers and sharing KV cache is new, as far as I know, although there are existing works on self-speculative decoding.
- The average accepted length and throughput achieved compare to Qwe 2.5 1.5B and Eagle-2 seems significant.

Weakness

- There doesn’t seem to be enough review of existing Self-speculative decoding methods, such as:
1. Draft on the Fly: Adaptive Self-Speculative Decoding using Cosine Similarity, EMNLP, 2024.
2. SWIFT: On-the-Fly Self-Speculative Decoding for LLM Inference Acceleration, ICLR, 2025.
3. Draft & Verify: Lossless Large Language Model Acceleration via Self-Speculative Decoding, ACL, 2024.
4. Kangaroo: Lossless Self-Speculative Decoding for Accelerating LLMs via Double Early Exiting, NeurIPS, 2024.
5. Generation Meets Verification: Accelerating Large Language Model Inference with Smart Parallel Auto-Correct Decoding, ACL, 2024.
- Not necessarily a weakness, but if SubSpec can be considered as a plug-in module for other self-speculative method, it would make a much stronger impact.  Specifically, would it be possible to combine layer-skipping [1,2] above and layer-quantization ideas? How compatible is SubSpec with existing self-speculative decoding methods?
- The statement of draft model being highly aligned doesn’t seem to be tested — although SubSpec achieves good end-to-end throughput improvement, would it be possible to showcase the output of draft and target are aligned at token level? perhaps a graph similar to Fig5 B in Eagle-2 paper would be helpful.
- Results are shown without standard variation or error bars, making it difficult to assess statistical significance.

---

> ### Author Rebuttal · Authors · 2025-07-30
>
> We thank the reviewer for the insightful feedback and questions. Your suggestions and questions have helped us clarify the novelty and practical considerations of our work. Please find our detailed responses below.
>
> ### 1. Missing Related Works
> Thank you for pointing this out. We will add a new subsection in our related works to provide a comprehensive review of self-speculative decoding methods:
> 1. Draft & Verify: Lossless Large Language Model Acceleration via Self-Speculative Decoding, ACL, 2024.
> 2. Kangaroo: Lossless Self-Speculative Decoding for Accelerating LLMs via Double Early Exiting, NeurIPS, 2024.
> 3. Generation Meets Verification: Accelerating Large Language Model Inference with Smart Parallel Auto-Correct Decoding, ACL, 2024.
> 4. Draft on the Fly: Adaptive Self-Speculative Decoding using Cosine Similarity, EMNLP, 2024.
> 5. SWIFT: On-the-Fly Self-Speculative Decoding for LLM Inference Acceleration, ICLR, 2025.
>
> We are willing to further expand this section in case we are missing important relevant works.
>
> ### 2. Compatibility with Self-SD Methods
> This is an excellent point. SubSpec is indeed designed as a modular framework and is highly compatible with other self-speculative methods like layer-skipping.
> The integration could be straightforward. For example, to combine SubSpec with layer-skipping, one could simply skip a subset of the offloaded layers entirely rather than replacing them with their quantized substitutes. The feasibility of this is supported by prior work (e.g., "Draft on the Fly," "SWIFT"), which shows high acceptance rates (α≈0.98) even when skipping 35-45% of the layers.
>
> Applying layer-skipping methods itself requires about 60% model VRAM usage, while a 4-bit substitute requires only 25%. A hybrid approach that combines layer-skipping and quantization could be even more powerful, potentially creating a draft model with a smaller VRAM footprint and faster generation speed than either method alone. We plan to investigate this direction further in our future work.
>
> ### 3. Proves On Token-Level Alignment
> While we are unable to generate a new figure during the rebuttal period, we can provide two forms of evidence for the high alignment between our draft and target models.
>
> **Existing Evidence in the Paper.** The most direct measure is the average token acceptance length (τ). As shown in Figure 2(a), SubSpec's τ is nearly twice as high as that of a small draft model of the same family, providing strong quantitative proof of superior token-level alignment.
>
> **Additional Experimental Evidence.**  We performed a simple experiment by computing the average cosine similarity of different draft models' output probability distributions against that of the target model, on each of the first four output tokens generated by the target model (Llama-3.1 8B, 5 samples in MT-Bench):
> | Draft Model | 1st token | 2nd token | 3rd token | 4th token |
> | :--- | :--- | :--- | :--- | :--- |
> | Llama-3.1 1B | 0.7965 | 0.5761 | 0.4718 | 0.2969 |
> | SubSpec | 0.9728 | 0.6012 | 0.5950 | 0.6038 |
>
> As shown, SubSpec maintains a much higher cosine similarity, confirming its superior token-level alignment compared to a smaller draft model of the same family.
>
> ### 4. Missing Standard Variation or Error Bars
> This is an important consideration. Due to the significant computational cost of our experiments, running multiple seeds for every configuration was infeasible.
>
> However, to validate the stability of our findings, we performed 5 independent runs on a key, representative result: SubSpec (8GB VRAM) on MT-Bench, obtaining a low standard deviation of 0.101 tokens/s.
> We will add this information and a discussion of it to the final version.
>
> ### Response to Questions
> > Could you please elaborate on how throughput translates to speedups? Especially Tab 1 is very similar to that of Eagle-2, but Eagle-2 reported speedups. It would be helpful to include speedups in the results.
>
> All of our experiments are conducted with a batch size of 1, as noted in Section 5 (Comparative Methodology and Parameters). Therefore, the speedup is simply the method's throughput divided by the offloading baseline's throughput.
> We chose to report absolute throughput primarily for transparency and fairness. We applied strong graph and kernel optimizations (torch.compile) to all methods, which significantly strengthened our baseline relative to those in other papers. Reporting only speedups against our optimized baseline could create an unfair comparison with previously published results.
>
> > Could you discuss the implementation complexity and system requirements? e.g., incorporating HQQ for quantization, and does the implementation of SubSpec pose any constraints on systems?
>
> The implementation of our method is straightforward, but its performance is dependent on the capabilities of the underlying quantization libraries, which introduce specific system constraints.
>
> **Implementation complexity.** The main implementation complexity lies in tree-based SD with shared KV-cache, which includes handling KV cache position indices correctly among draft and target models, during both decoding and verification processes. The incorporation of HQQ for quantization and asynchronous offloading only adds minimal implementation complexity.
>
> **System requirements and constraints.**  The primary system constraint is the need for quantization kernels with efficient GEMM support. Tree-based speculative decoding processes multiple tokens in parallel, requiring matrix-matrix (GEMM) operations. Many low-bit (especially <4 bits) quantization libraries are only optimized for single-token autoregressive generation, which only relies on matrix-vector (GEMV) operations. This technical requirement is why our work focuses on HQQ with 4-bit quantization, as it is one of the few libraries with performant GEMM support for quantized models. We anticipate that broader low-bit GEMM support will become available as (tree-based) speculative decoding grows in popularity.

---

> > ### Comment · Reviewer_DA4c · 2025-08-05
> >
> > Thank you for your response to my review comments. My concerns have been sufficiently addressed, and I am raising my score to Accept.

---

> > > ### Author Response · Authors · 2025-08-08
> > >
> > > Thank you again for your insightful feedback. We are already working on revising the paper accordingly.

---

### Official Review · Reviewer_i2M6 · 2025-07-03

**Clarity:** 3
**Significance:** 2
**Originality:** 2
**Rating:** 4
**Confidence:** 3

**Summary:**

The paper proposed a technique for speculative decoding in offloading setting, where the inference is completely memory bound. The paper first identified that offloading workload requires different draft model from Eagle, where the draft model less needed to be fast. For offloading, the draft model needs to have strong performance in drafting deep and accurate trees for expensive offloading full model verification. The paper proposes a method for speculative decoding in offloading, where it used the quantized model as the draft model that can fit inside the consumer GPU VRAM. Model weights and KV cache are shared between draft and the full model.

**Questions:**

Under the same budget, quantized model might not have significant tree building capacity compared to using a small draft model and sample a huge tree. However, I think it really depends on the target hardware and the full model parameter count. I think there is some value in studying when it is better to use compressed model of the full as the draft model, when using a smaller draft model is more favorable.

**Ethical Concerns:**

["NO or VERY MINOR ethics concerns only"]

**Final Justification:**

My two concerns remain.
1. SpecExec comparison is critical in establishing the conclusion of the paper. The current experiment results requires important clarification in the experiment setup, which is important for establishing the comparison.
2. The method fundamental difficulty to extend to the large model regime also raises questions about the scope of the paper.

Based on the above two reasons, my concern stands. Therefore, I will not raise the score.

**Limitations:**

Yes

**Paper Formatting Concerns:**

No obvious formatting issue found.

**Quality:**

3

**Strengths And Weaknesses:**

Strengths
1. The paper is well-written and easy to follow
2. The contrasts between offloading speculative decoding with normal SD is clear and insightful
3. The paper show practical speedup, reinforcing the usefulness of the proposed method

Weaknesses
1. Important comparison missing, need to compare against SpecExec, as it is mentioned in the related work section, and it is clearly a much better baseline than Qwen Eagle
2. The proposed method might encounter diminishing benefit when tackling 70B or beyond models offloading on constraint devices, as the quantized model might not even fit on the on-chip memory, and you have to resort to a non weight sharing draft model that is smaller (8B model).
3. Minor, lack of discussion on prior self-speculative decoding works.
Elhoushi, M., Shrivastava, A., Liskovich, D., Hosmer, B., Wasti, B., Lai, L., ... & Wu, C. J. (2024). LayerSkip: Enabling early exit inference and self-speculative decoding. arXiv preprint arXiv:2404.16710.
Zhou, Y., Chen, Z., Xu, Z., Lin, V., & Chen, B. (2024). Sirius: Contextual sparsity with correction for efficient llms. arXiv preprint arXiv:2409.03856.

---

> ### Author Rebuttal · Authors · 2025-07-30
>
> We thank the reviewer for the detailed and constructive feedback. We have conducted new experiments based on your suggestions, which we believe have strengthened the paper. Please find our responses below:
>
> ### 1. Missing Direct Comparison with SpecExec
> We agree that a direct comparison with SpecExec is a crucial baseline. While we previously included a discussion in Appendix B.2, we have now performed a direct performance comparison. Under a similar decoding budget (~288 nodes), SubSpec significantly outperforms SpecExec:
> | Method | Throughput (tokens/s) | τ |
> | :--- | :--- | :--- |
> | **SubSpec (k6-d48)** | **24.29** | **29.66** |
> | SpecExec (k = 32, budget = 288) | 13.253 | 11.23 |
>
> SubSpec achieves ~83% higher throughput, driven by a 2.6x higher average token acceptance length (τ). This result validates SubSpec’s design principle: using a highly-aligned draft model combined with system co-design is more effective. We will prominently feature this direct comparison in the revised Appendix B.2.
>
> ### 2. Diminishing Benefit When Scaling to Very Large Models
> This is a valid point regarding the method's scope. Our approach is primarily designed for the common and practical scenario where a quantized version of the target model can fit into GPU VRAM. This covers a wide range of models (up to the 30B scale) on modern consumer GPUs.
> For extreme cases, such as running a 70B+ model on a single constrained device where even the quantized draft will not fit, a different approach (e.g., using a separate, smaller draft model) would indeed be necessary.
> However, SubSpec remains applicable for these large models when multiple GPUs are available (e.g., via tensor parallelism), offering an efficient and training-free alternative where training a custom draft model might be infeasible.
>
> ### 3. Missing Related Works
> Thank you for pointing this out. We will add a new subsection in our related works to provide a comprehensive review of self-speculative decoding methods.
>
> ### Response to Questions
> > Under the same budget, quantized model might not have significant tree building capacity compared to using a small draft model and sample a huge tree ... I think there is some value in studying when it is better to use compressed model of the full as the draft model, when using a smaller draft model is more favorable.
>
> Thanks for this insightful question. We investigated this trade-off by testing whether a larger draft tree could compensate for a smaller draft model. We expanded the tree width (k) from 6 to 32 and observed the following:
> | Draft Model | Throughput (tokens/s) | τ |
> | :--- | :--- | :--- |
> | **SubSpec (k6-d48)** | **24.29** | **29.66** |
> | Llama-3.1 1B (k6-d32) | 15.14 | 11.91 |
> | Llama-3.1 1B (k32-d32) | 13.77 | 15.41 |
>
> Our results show that increasing the tree width yields minor gains in average token acceptance length (τ), while harming throughput due to the increased drafting overhead. We attribute this to the highly skewed probability distributions of LLMs, where a small number of top tokens account for the vast majority of the probability mass.
>
> This finding suggests that SubSpec will always be more favorable against less-aligned small models of the family while sampling huge draft trees. We will add this valuable analysis to the final paper.
> Thank you again for helping us improve our work.

---

> > ### Comment · Reviewer_i2M6 · 2025-08-06
> > **Response**
> >
> > Thank you for your time and effort in commenting on my review.
> > 1. Some questions about the SpecExec Comparison
> > SpecExec presents 70B model scale for their offloading experiments. What is the setting here? What is the model size, and what is the dataset used to evaluate the experiments? Also, since SpecExec may not use the quantized model as the draft, but resort to a much smaller model as the draft. Therefore, the number of nodes in the decoding tree isn't a valid control for the experiments; instead, flops control might be a much more reasonable design. The above questions are critical to understand the comparison with SpecExec.
> > 2. Larger models
> > I appreciate the author's clarification. I still think larger models are also very important for Speculative Decoding methods. However, I do agree with the considerations presented in the review, but my doubt stands.
> > Based on the above concerns, without further clarification, I maintain my score.

---

> ### Author Response · Authors · 2025-08-06
>
> Thank you again for your constructive comments and for pushing us to clarify this critical comparison.
>
> ### Reply to Point 1 (Fair Comparison & VRAM Constraints)
> All experiments in this discussion thread use Llama-3.1 8B as the target model and Llama-3.2 1B as the draft model, running on MT-Bench (greedy decoding, 8GB VRAM).
>
> You are correct that controlling for the number of tree nodes is not ideal when draft models differ. The ideal approach is to find the maximal achievable throughput for each method under the same hardware constraint. However, there is a critical issue when attempting this comparison: the official SpecExec implementation fails to stay within the 8GB VRAM limit, even with small tree sizes and a reduced KV cache of 1024 (which is 2048 in our paper).
> To find SpecExec's best possible performance regardless of this constraint, we still conducted a full parameter sweep of SpecExec’s budget (its parameter for draft tree size):
> | Budget | Used VRAM (GB) | Throughput (tokens/s) | τ |
> | :--- | :--- | :--- | :--- |
> | 64 | 8.38 |11.75 | 8.64 |
> | 128 | 8.43 |12.21 | 9.42 |
> | 256 | 8.55 |13.02 | 10.79 |
> | 512 | 8.79 | 13.77 | 12.40 |
> | 1024 | 9.27 | 12.95 | 12.93 |
> | 2048 | 10.41 | 11.61 | 14.12 |
> | 4096 | 15.00 | 6.40 | 14.53 |
>
> SpecExec's peak throughput is 13.77 tokens/s (at a budget of 512). This is much lower than SubSpec's 24.29 tokens/s, and is achieved while already exceeding the 8GB VRAM limit.
> This comprehensive sweep also reinforces our earlier conclusion: the strategy of using a smaller draft model with a very large (width) tree may yield diminishing returns. Beyond a certain point (budget=512), the overhead of processing the large draft tree during the drafting phase outweighs the gains in acceptance length, causing overall throughput to decrease.
>
> ### Reply to Point 2 (Scalability to Large Models)
> We agree that scalability to very large models is an important consideration.
> Although we have already addressed this limitation in Appendix A of our submission, we will incorporate the discussion above into this section.

---

### Official Review · Reviewer_fdZL · 2025-07-03

**Clarity:** 2
**Significance:** 3
**Originality:** 3
**Rating:** 4
**Confidence:** 3

**Summary:**

The authors propose SubSpec, an effective speculative decoding method designed for settings where parameter offloading is required. The core idea is to retain a low-bit quantized replica of the model on the GPU as a draft model, while periodically loading the full-precision parameters of the original model to perform lossless speculative decoding. Importantly, this approach does not require any additional training. In addition, the authors introduce several optimizations, such as modifications to the draft tree construction strategy and asynchronous execution, to further improve decoding speed.

**Questions:**

1. Could the authors provide experimental results comparing different quantization algorithms and target bit-widths?

2. Although admittedly challenging, could the authors consider providing a direct comparison with modern quantized models?

**Ethical Concerns:**

["NO or VERY MINOR ethics concerns only"]

**Final Justification:**

The authors have largely addressed the concerns I raised. Although the resulting trade-off, achieving a lossless 4-bit quantized model at the cost of roughly 50% efficiency, may be viewed differently depending on one’s perspective, I find the manuscript overall to be sufficiently coherent and complete.

**Limitations:**

yes

**Paper Formatting Concerns:**

No major concerns

**Quality:**

3

**Strengths And Weaknesses:**

**Strengths**

1. The authors present a clear and practical use case, i.e. deploying larger-scale language models on mainstream consumer GPUs through hybrid optimization strategies.

2. The experimental evaluation is thorough. The authors provide detailed deployment data, instilling confidence in reproducibility. They validate the method across models of varying sizes and settings, demonstrating its broad applicability. The reported improvements are substantial, with speedups of up to 30× in some scenarios.

3. The authors keenly identify a key limitation in applying traditional draft tree strategies to SubSpec: SubSpec employs a significantly more capable draft model compared to traditional speculative decoding setups, where the draft model is relatively weak. As a result, conventional draft tree construction strategies may not be optimal. The authors propose a revised strategy accordingly, offering valuable insight for future draft tree designs.

4. The paper includes ablation studies to quantify the contributions of different components to overall performance gains.

**Weaknesses**

The discussion of quantization strategies and target bit-widths is somewhat limited. Comparative experiments in this area would strengthen the completeness of the study.

The primary weakness of the paper lies in the narrow interpretive framing of SubSpec. The method is discussed solely from the speculative decoding perspective, as an offload-aware approach to improving inference efficiency. However, SubSpec can also be viewed through the lens of quantization compensation: a low-bit quantized model is augmented by periodically verifying its predictions using a full-precision model, effectively correcting potential inference errors. From this viewpoint, the appropriate baselines include not only other speculative decoding methods, but also standard quantization techniques.

To fairly assess SubSpec's tradeoffs, it would be necessary to compare both the efficiency and capability of SubSpec against strong quantized baselines. Although SubSpec aims to be training-free, high-performance quantized models, especially those using state-of-the-art quantization algorithms, are often readily available. Furthermore, recent 4-bit quantization techniques (as used in this paper) tend to incur minimal accuracy degradation while delivering significant throughput gains by eliminating the need for offloading entirely. It remains unclear whether SubSpec offers a favorable efficiency–capability tradeoff, or if it sacrifices excessive efficiency for relatively minor capability gains.

---

> ### Author Rebuttal · Authors · 2025-07-30
>
> We thank the reviewer for the insightful feedback.
>
> We agree that viewing SubSpec as a form of quantization compensation is a valuable perspective that helps clarify its unique contribution.
> The core design principle of SubSpec is to achieve acceleration while being lossless, guaranteeing outputs that are bit-for-bit identical to the full-precision (e.g., fp16) model. This fundamentally distinguishes our method from standard quantization techniques, which are inherently lossy and inevitably alter model outputs. This distinction is critical for applications where any deviation from the original model's behavior is unacceptable.
>
> While we acknowledge that modern 4-bit quantization techniques achieve only 1~2% accuracy degradation on multiple tasks [1,2], the accuracy degradation is not always negligible, especially on more challenging benchmarks [2]. To illustrate this, we evaluated several popular 4-bit quantization methods on MMLU-Pro [3], using a random sample of 50 questions per category. The results showed a non-trivial accuracy drop in accuracy compared to the fp16 baseline:
> | Model Configuration | Accuracy (%) |
> | :--- | :--- |
> | Full Precision (fp16) | 46.1 |
> | GPTQ (int4) | 41.8 |
> | AWQ (int4) | 44.1 |
> | HQQ (int4) | 42.2 |
>
> SubSpec avoids this accuracy-capability trade-off entirely. Nonetheless, we agree that a direct comparison against strong quantized baselines is essential. Hence, we now include a direct performance comparison against fully GPU-resident 4-bit models to provide a clearer picture of the efficiency-capability landscape. The table below shows the throughput of Llama-3.1 8B on a single GPU with an 8GB VRAM limit on MT-Bench:
> | Model Configuration | Throughput (tokens/s) |
> | :--- | :--- |
> | Full Precision (fp16) | 2.40 |
> | GPTQ (int4) | 58.87 |
> | AWQ (int4) | 52.32 |
> | HQQ (int4) | 135.84 |
> | **SubSpec** | 24.29 |
>
> While a standalone 4-bit model offers maximum throughput by accepting the quality trade-offs of quantization, SubSpec is designed for users who require lossless output that is bit-for-bit identical to the original FP16 model.
>
> For this high-fidelity use case, SubSpec makes offloading practical. It accelerates naive offloading from an unusable 2.4 tokens/sec to an acceptable 24 tokens/sec for interactive use. This allows users to achieve full model fidelity at a speed that is comparable to some standalone quantized models, all on consumer-grade hardware. We therefore position SubSpec as a simple, training-free solution for users who cannot compromise on model quality.
>
> ### Response to Questions
> > Could the authors provide experimental results comparing different quantization algorithms and target bit-widths?
>
> **Different quantization algorithms.**
> We have shown speed results of different quantization methods above.
> The reason we chose HQQ (int4) for our experiments is primarily due to its training-free nature and its high-performance triton kernels, which support efficient GEMM operations essential for tree-based speculative decoding.
>
> **Different target bit-widths.**
> While state-of-the-art low-bit methods like QUIP# [4] and QTIPS [5] (2-bit) show promising results, their current public kernels lack GEMM support, making them incompatible with tree-based speculative decoding methods. We thus consider exploring draft models with lower bitwidths (2/3 bits) as promising future work, contingent on kernel development. Please refer to our reply for Reviewer DA4c for more details about the system requirements.
>
> We have also added experiments using SubSpec to accelerate an 8-bit target model to provide a more comprehensive analysis.
> This scenario is considered as a different practical trade-off, where the accuracy degradation introduced by 8-bit quantization is acceptable.
>
> Considering running Llama-3.1 8B on MT-Bench under an 8GB VRAM limit, while quantizing the target model to 8-bit still requires offloading, its memory footprint and offloading time are nearly halved compared to the 16-bit version.
> Our results show this configuration increases throughput by 36%, presenting a compelling trade-off between precision and performance:
> | Configuration (Target, Draft) | Throughput (tokens/s) | τ |
> | :--- | :--- | :--- |
> | SubSpec (fp16, int4) | 24.29 | 29.66 |
> | SubSpec (int8, int4) | 33.029 | 29.764 |
>
> > Although admittedly challenging, could the authors consider providing a direct comparison with modern quantized models?
>
> Yes. We have compared the throughputs of different quantized models and discussed trade-offs between SubSpec and strong quantized baselines above, hoping to address your concerns.
>
> #### Reference
> 1. An Empirical Study of LLaMA3 Quantization: From LLMs to MLLMs, Visual Intelligence, 2024
> 2. Quantization Hurts Reasoning? An Empirical Study on Quantized Reasoning Models, COLM, 2025
> 3. MMLU-Pro: A More Robust and Challenging Multi-Task Language Understanding Benchmark, NeurIPS, 2024
> 4. QuIP#: even better LLM quantization with hadamard incoherence and lattice codebooks, ICML, 2024
> 5. QTIP: Quantization with Trellises and Incoherence Processing, NeurIPS, 2024

---

> ### Comment · Reviewer_fdZL · 2025-08-06
>
> Thank you to the authors for the comprehensive response. I believe the main concern I raised—specifically, the comparison with quantization methods—has been adequately addressed. In addition, the experiments under different precision settings serve as a valuable supplement. The key logical issues I previously noted in the manuscript have been resolved. I will adjust my score accordingly, with the specific changes to be determined upon further consideration. I wish the authors all the best.

---

> ### Author Response · Authors · 2025-08-08
>
> Thank you again for your valuable feedback. We are already working on revising the paper accordingly.

---

### Official Review · Reviewer_BRmM · 2025-07-05

**Clarity:** 3
**Significance:** 2
**Originality:** 3
**Rating:** 3
**Confidence:** 4

**Summary:**

This paper addresses the significant inference latency of large language models (LLMs) when using parameter offloading to run them on consumer GPUs with limited VRAM. The authors propose Substitute Speculative Decoding (SubSpec), a novel, training-free method to accelerate these offloaded models.

The core contribution is a technique to construct a highly-aligned, GPU-resident draft model for speculative decoding. This is achieved by creating low-bit quantized 'substitute' layers for the offloaded parts of the target LLM, while directly sharing the layers that already fit in GPU memory as well as the KV-Cache. This design ensures high alignment between the draft and target models without requiring any training. The paper also introduces refinements to the draft tree construction, such as probability sharpening, to maximize the token acceptance rate.

**Questions:**

Please refer to the weakness part.

**Ethical Concerns:**

["NO or VERY MINOR ethics concerns only"]

**Limitations:**

Yes

**Quality:**

3

**Strengths And Weaknesses:**

Strength:

* The paper is exceptionally well-written and clearly structured.
* The central idea of using a low-bit quantized version of the target LLM as the draft model is both intuitive and highly effective.
* The paper demonstrates outstanding empirical results, particularly under the greedy decoding setting (temperature = 0).

Weakness:

The primary weakness of the paper lies in its experimental comparison framework, which may not fully address the most common and practical trade-offs for deploying LLMs on consumer hardware.
* Missing Comparison with Standalone Quantized Models: The paper frames SubSpec as a solution to the high latency of offloading half-precision models. However, a very common and effective strategy for memory-constrained devices is to use a fully quantized model (e.g., 4-bit) that fits entirely within GPU VRAM, thus avoiding the offloading bottleneck from the start. The quality loss from modern quantization techniques is often considered an acceptable trade-off for the immense speed gain. The paper lacks a direct performance comparison against this strong and highly relevant baseline. It is unclear how SubSpec's performance (e.g., 25 tokens/s for Qwen2.5 7B) compares to simply running a 4-bit quantized version of the same model directly on the GPU.
* Untapped Potential in "Quantized-to-Quantized" Scenarios: Building on the previous point, a more practical application of speculative decoding in resource-limited settings would be to accelerate an already-quantized target model. For example, one could use a highly compressed model (e.g., 2-bit or 3-bit) as a draft for a less compressed, higher-fidelity target model (e.g., 4-bit or 5-bit) that is also fully GPU-resident. This "quantized-to-quantized" speculative decoding represents a more realistic optimization scenario for end-users. The paper does not explore this direction.

Without this analysis, the experimental comparison is incomplete. Demonstrating a clear and favorable trade-off in this more practical context would significantly strengthen the paper's contribution. Addressing this limitation by providing a more comprehensive comparison against strong, quantized baselines would make the work's practical utility much clearer, and I would be inclined to raise my score accordingly.

---

> ### Author Rebuttal · Authors · 2025-07-30
>
> We thank the reviewer for the very thoughtful and crucial review. We will address the weaknesses in the following:
>
> ### 1. Missing Comparison with Standalone Quantized Models
> We agree that comparing SubSpec to a standalone quantized model is essential. As you suggested, we now include a direct performance comparison against fully GPU-resident 4-bit models.
>
> The table below shows the throughput of Llama-3.1 8B on a single GPU with an 8GB VRAM limit:
> | Model Configuration | Throughput (tokens/s) |
> | :--- | :--- |
> | Full Precision (fp16) | 2.40 |
> | GPTQ (int4) | 58.87 |
> | AWQ (int4) | 52.32 |
> | HQQ (int4) | 135.84 |
> | **SubSpec** | 24.29 |
>
> While a standalone 4-bit model offers maximum throughput by accepting the quality trade-offs of quantization, SubSpec is designed for users who require lossless output that is bit-for-bit identical to the original FP16 model.
>
> For this high-fidelity use case, SubSpec makes offloading practical. It accelerates naive offloading from an unusable 2.4 tokens/sec to an acceptable 24 tokens/sec for interactive use. This allows users to achieve full model fidelity at a speed that is comparable to some standalone quantized models, all on consumer-grade hardware. We therefore position SubSpec as a simple, training-free solution for users who cannot compromise on model quality.
>
> ### 2. Untapped Potential in Quantized-to-Quantized Fully GPU-resident Scenarios
> We agree it is an interesting direction. However, for a draft with 2-bit substitutes accelerating a 4-bit target, we believe the gains would be modest. A much more efficient draft architecture (e.g., EAGLE) is required to achieve decent speedup on fully GPU-resident cases, as shown in our analysis in Sections 3.1 and 3.2. The effectiveness of speculative decoding hinges on a draft model that is substantially faster than the target. Also, due to hardware and kernel overheads, current 2-bit quantization kernels are not yet fast enough compared to their 4-bit counterparts to create the large speed gap needed for significant improvement.
>
> Inspired by this suggestion, we analyzed a different practical trade-off, where the accuracy degradation introduced by 8-bit quantization is acceptable.
>
> Considering running Llama-3.1 8B on MT-Bench under an 8GB VRAM limit, while quantizing the target model to 8-bit still requires offloading, its memory footprint and offloading time are nearly halved compared to the 16-bit version. Our results show this configuration increases throughput by 36%, presenting a compelling trade-off between precision and performance:
> | Configuration (Target, Draft) | Throughput (tokens/s) | τ |
> | :--- | :--- | :--- |
> | SubSpec (fp16, int4) | 24.29 | 29.66 |
> | SubSpec (int8, int4) | 33.029 | 29.764 |
>
> Thank you again for pushing us in these valuable directions. We will incorporate this analysis into the final paper.

---

> > ### Author Response · Authors · 2025-08-08
> >
> > Thank you again for your thoughtful comments.
> >
> > Since your review, we have added comparisons with standalone quantization methods and addressed the difference between our method and running standalone quantization methods directly: we position SubSpec as a simple, training-free solution for users who cannot compromise on model quality, while standalone quantization methods on the other hand, inevitably alter model outputs.
> >
> > For the potential quantized-to-quantized scenario, we analyzed a different practical trade-off: accelerating quantized 8-bit target model with SubSpec, resulting in a higher throughput if users slight accuracy degradation is acceptable.
> > Do these updates address your doubts? As the discussion phase is nearing an end, we would really appreciate any feedback or address additional questions before the deadline.

---

### Note · Authors · 2025-08-12

Dear Reviewers and AC,

We sincerely thank you for your constructive feedback. We have conducted several new experiments and analyses to address the key concerns raised during the review period. Here is a summary of the major updates:
### 1. Discussion on Comparison with Standalone Quantization Methods.
According to reviewer suggestions (BRmM, fdZL), we added direct comparisons against fully GPU-resident 4-bit models. We clarified SubSpec's role as a *bit-for-bit* solution for users who require output identical to the original FP16 model. For this high-fidelity use case, SubSpec boosts naive offloading from an unusable **2.4 tokens/s** to a practical **24 tokens/s** for interactive use.

### 2. Discussion on Quantized-to-Quantized Scenarios.
Inspired by reviewer BRmM’s feedback, we tested a new INT8 target + INT4 draft scenario. This practical configuration increased throughput by **36%** compared to the FP16 target, demonstrating a compelling new performance trade-off.

### 3. Discussion on Comparison with SpecExec Under Same VRAM Limits
Reviewer i2M6 raised concerns about not directly comparing with SpecExec.
While we have already included a discussion in **Appendix B.2**, we further conducted a full parameter sweep to find its peak throughput under 8GB VRAM constraint for fair comparison. The results are shown in the table of our reply under
**"Reply to Point 1 (Fair Comparison & VRAM Constraints)"**.

Our analysis revealed two key findings:
- The official SpecExec implementation fails to stay within the 8GB VRAM limit, even with halved KV-Cache.
- Even when allowing it to exceed the memory limit, SpecExec's peak throughput (**13.77 tokens/s**) is significantly lower than SubSpec's (**24.29 tokens/s**).

This new data provides a more rigorous comparison and reinforces our paper's conclusions about the benefits of a highly-aligned draft model.
From the results, we also addressed the reviewer’s question by showing that increasing the size of the draft tree (budget size) beyond a certain point may cause the overall throughput to decrease.

### 4. Discussion on Self-Speculative Decoding Related Works
Reviewer i2M6 and DA4c noted the need to discuss related works about self-speculative decoding.
We will add an extra section about the related works in the Appendix.

---

### Decision · Program_Chairs · 2025-09-17

**Decision:**

Accept (poster)

**Comment:**

(a) Summary

The paper proposes SubSpec which is a new speculative decoding method for offloaded LLMs on consumer GPUs. It builds a quantized substitute draft model from the target’s offloaded layers, shares GPU-resident layers and KV cache, and adapts draft-tree construction for high alignment.

(b) Strengths

- Addresses a practical deployment bottleneck with a simple, effective design.
- Strong empirical results:  speedup, alignment.
- Added key comparisons: 4-bit baselines, INT8+INT4 variant, SpecExec sweep under VRAM limits.

(c) Weaknesses

- No FLOPs-controlled fairness in SpecExec comparison (though VRAM-constrained sweep provided).
- Limited discussion of very large models (70B+)—acknowledged as out of scope.
- Related work on self-speculative decoding initially thin (to be added).

(d) Reasons

The method is well-motivated, practical, and now thoroughly evaluated. Authors addressed major concerns with new experiments and clarifications. Remaining issues are minor and suitable for camera-ready.

(e) Discussion & Rebuttal

Reviewers asked for quantized baselines, quantized-to-quantized scenarios, SpecExec comparison, and alignment evidence. Authors delivered all, plus clarified scope and added stability data. One reviewer (BRmM) didn't engage in the post-rebuttal discussion while mentioning he/she would adjust the score after the discussion. So, this review needed to be down-weighed. One reviewer maintained a borderline score citing FLOPs fairness, but this is a caveat, not a blocker. Overall, the concensus is that the rebuttal addressed major concerns.